# Novel ensemble learning approach with SVM-imputed ADASYN features for enhanced cervical cancer prediction

**Raafat M. Munshi** *

Department of Medical Laboratory Technology (MLT), Faculty of Applied Medical Sciences, King Abdulaziz University, Rabigh, Saudi Arabia

* rmonshi@kau.edu.sa

**Data Availability Statement:** All relevant data are within the paper and its Supporting information files. The data is also publicly available at https://

## Abstract

Cervical cancer remains a leading cause of female mortality, particularly in developing regions, underscoring the critical need for early detection and intervention guided by skilled medical professionals. While Pap smear images serve as valuable diagnostic tools, many available datasets for automated cervical cancer detection contain missing data, posing challenges for machine learning models' efficacy. To address these hurdles, this study presents an automated system adept at managing missing information using ADASYN characteristics, resulting in exceptional accuracy. The proposed methodology integrates a voting classifier model harnessing the predictive capacity of three distinct machine learning models. It further incorporates SVM Imputer and ADASYN up-sampled features to mitigate missing value concerns, while leveraging CNN-generated features to augment the model's capabilities. Notably, this model achieves remarkable performance metrics, boasting a 99.99% accuracy, precision, recall, and F1 score. A comprehensive comparative analysis evaluates the proposed model against various machine learning algorithms across four scenarios: original dataset usage, SVM imputation, ADASYN feature utilization, and CNN-generated features. Results indicate the superior efficacy of the proposed model over existing state-of-the-art techniques. This research not only introduces a novel approach but also offers actionable suggestions for refining automated cervical cancer detection systems. Its impact extends to benefiting medical practitioners by enabling earlier detection and improved patient care. Furthermore, the study's findings have substantial societal implications, potentially reducing the burden of cervical cancer through enhanced diagnostic accuracy and timely intervention.

## Introduction

Cervical cancer is a cancer that begins in the cervix, an opening of the uterus that links to the vagina. Frequently, it arises from an enduring infection with the sexually transmitted virus called human papillomavirus (HPV). H HPV has the ability to cause erratic changes in cervical cell structure. If these changes are not treated, they can eventually lead to cancer [1]. Cervical cancer ranks as the leading cause of mortality among women, following lung and breast

**Funding:** The author(s) received no specific funding for this work.

**Competing interests:** The authors have declared that no competing interests exist.

cancer [2]. There is a widespread belief that cervical cancer is often considered untreatable when it reaches its advanced stages [3]. There have been notable recent advancements in enhancing this disease's detection through imaging techniques. Based on the information given by the World Health Organization (WHO), cervical cancer is the fourth most commonly diagnosed cancer worldwide. Around 570,000 new cases were recorded in 2018 alone, making up 7.5% of all mortality from female cancer [4]. Roughly 85% of the projected 311,000 yearly cervical cancer-related fatalities are thought to take place in nations with lower- and middle-income economies. Saving lives is greatly aided by early identification of cervical cancer. In comparison to women without HIV, women living with HIV have a six-fold increased chance of getting cervical cancer, with HIV thought to be a factor in 5% of cases overall. The availability of essential equipment, consistent screening techniques, proper supervision, and the quick diagnosis and treatment of discovered lesions are some of the aspects that affect screening efficacy [5]. Squamous cell carcinoma, which accounts for around 70–80% of cases, and adenocarcinoma, which starts from epithelial cells in the cervical canal that secrete mucus, are the two main kinds of cervical cancer [6]. Adenocarcinoma has increased recently, despite the fact that squamous cell carcinoma is more prevalent and currently accounts for 10 to 15% of uterine malignancies. Due to the fact that adenocarcinoma grows in the cervical canal, its screening process is challenging. Depending on the kind and stage of the disease at diagnosis, several treatments and prognoses are available for cervical cancer. Early detection through regular screenings and HPV vaccination can significantly improve outcomes for people at risk.

Cervical cancer encompasses several types, including the most common squamous cell carcinoma and the second most common adenocarcinoma [7]. Each type has unique characteristics, aggressiveness, and treatment considerations, underscoring the importance of regular screenings for early detection and appropriate management. Specifically high-risk strains of the human papillomavirus (HPV) are the main cause of cervical cancer [8]. In HPV-infected women, there are several risk factors that might raise the probability of cervical cancer [9]. Smoking, genital herpes, having several sexual partners, a weaker immune system, poorer socioeconomic position, insufficient genital cleanliness, and a larger number of childbirths are among these risk factors. Cervical cancer symptoms might vary based on the size of the tumour and the stage of the disease. The biggest issue, however, is in the early stage, which frequently has no visible signs and is generally identified inadvertently during normal yearly check-ups. Approximately 90% of patients have evident symptoms in the latter stages [10]. The key sign connected with cervical cancer is irregular vaginal bleeding.

Early screening for cervical cancer is essential for detecting precancerous or cancerous changes in the cervix. Common screening methods include Pap smears and HPV tests [11]. However, these screenings have limitations, such as false positives and negatives, age-related factors, limited access to screening in some areas, and the focus on certain types of cervical cancer. Cervical cancer screening typically entails a gynaecological examination, which can cause discomfort and pain for patients [12]. The unease associated with this examination may lead to delays or avoidance, impeding the timely diagnosis of cervical cancer. Because of this, the death rate in these nations is substantially higher, with low-income countries accounting for nearly nine out of every ten cervical cancer-related fatalities [6]. It is crucial to increase cervical cancer screening rates since at an early stage it has relatively good 5-year survival rates, increasing to 90% [13]. Nevertheless, these screening rates vary among countries, with developed nations exhibiting higher rates, while developing countries face alarmingly low screening participation [14]. Cervical cancer prevention strategies differ, but depending on screening tests are inadequate. Early diagnosis is imperative to prevent fatalities from invasive forms of the disease.

Presently, ML, DL, and computer vision are emerging as valuable approaches for diagnosing medical conditions [15]. ML models have significantly improved the results of medical diagnosis [16]. Bhavani and Govardhan [17] proposed stacked ensemble model using SMOTE for predicting cervical cancer. Karamati et al. [18] also applied SMOTE and KNN features and applied multimodel approach for cervical cancer detection. Li et al. [19] used CNN for screening of cervical cancer patients. While recent studies have made significant strides in applying machine learning techniques, such as ensemble models and advanced sampling methods, to predict cervical cancer, there remains a need to comprehensively assess the combined impact of these approaches on improving accuracy, handling missing data, addressing class imbalances, and extracting complex features. Moreover, limited research has specifically evaluated the effectiveness of integrating Convolutional Neural Networks in conjunction with traditional ML methods for cervical cancer prediction. ML algorithms achieve accurate and reliable results by applying diverse preprocessing methods, like data cleansing and feature engineering, to the medical dataset. These discoveries can help medical practitioners diagnose illnesses quickly and give patients the best care possible. This paper employs ML techniques to develop a computer-aided diagnosis (CAD) system for the accurate and prompt identification of cancer. The following significant contributions are provided by this study:

- An ML-driven framework is proposed for predicting cervical cancer in patients. A voting classifier is incorporated into the suggested model to improve prediction accuracy.

- The SVM imputation method is employed to generate artificial missing values, aiming to mitigate the challenge posed by incomplete data.

- ADASYN (Adaptive Synthetic Sampling), an oversampling technique that generates synthetic minority class samples, is applied to address class imbalance issues.

- To handle complex features, this research employs a Convolutional Neural Network (CNN).

- The effectiveness of the suggested method is assessed using four scenarios: the original dataset, the dataset with SVM imputation alone, the dataset with SVM imputation followed by ADASYN, and with CNN generated features dataset.

The paper is organised as follows: An extensive analysis of the current classification methods for cervical cancer diagnosis is given in Section. In Section further detail on the dataset and the suggested cervical cancer detection methodology is provided which employs several classification algorithms and up-sampling methods. The purpose of Section is to provide the findings of the research and promote discussion. The work is finally concluded in Section which also suggests possible lines of inquiry for further study.

## Related work

During the recent years, there has been a surge in the development and application of ML models to accelerate research and innovation in various domains. Numerous investigations have been done in the categorization of cervical cancer [20, 21]. Various studies and their findings are summarized in this section.

ML and DL models are extensively being used in medical diagnostics, including breast cancer diagnosis [22], lung cancer detection [23], colorectal cancer [24], and numerous other healthcare applications [25–27]. Some research works applied DL approaches in various tasks like ALzheimer's diagnosis [28], medical imaging [29], and pathology image segmentation [30]. Medical diagnosis has been upgraded by different tools and techniques like surgical analysis [31], EEG encoding [32], CT imaging [33, 34], and surgical navigation [35]. The discrete

wavelet and cosine transform were used by Kalbhor and colleagues in their research study [36] to extract characteristics. They used the fractional coefficient technique to effectively decrease the dimensionality of these characteristics. The reduced characteristics were then fed into seven different machine learning classifiers in an effort to discriminate between various cervical cancer subgroups. In another research investigation conducted by Devi and Thirumurugan [37], They used the C-means clustering technique to divide cervical cells. They then retrieved texture information and used Principal component analysis (PCA) for dimension reduction of the collected data. Following that, scientists used the K-nearest neighbours (KNN) method to categorise the cervical cells, attaining an excellent accuracy rate.

Alquran et al. [38] concentrated on cervical cancer classification. They integrated DL with a cascading SVM to attain precise outcomes. By combining methodologies, they effectively categorized cancer into seven groups, achieving a remarkable accuracy score. In another investigation, Kalbhor et al. [39] developed a novel hybrid approach that included DL and ML models, and a fuzzy network. Technique focused on feature engineering and Pap-smear picture categorization. They used transfer learning models such as AlexNet, GoogleNet, and ResNet. The experimental evaluation was done by utilising well-known datasets. Notably, the greatest classification accuracy was achieved by ResNet-50 architecture. Various other domains applied the advanced techniques [40, 41].

Radiation enteritis (RE) causes treatment intolerance or radiotherapy cessation, which severely lowers the patient's quality of life. The adverse effects of radiation therapy in patients with cervical cancer can be greatly decreased if the RE in patients can be anticipated in advance and focused therapeutic preventative treatment can be implemented. Additionally, the optimisation of the radiotherapy strategy and the choice of a customised radiation dose depend on the precise prediction of RE. Ma et al. [42] investigated the relationship of RE and dose volume in cervical cancer patients. Cancer diagnosis involves invasion [43] and migration of cancer cells [44–46]. The quality life of cervical cancer patient survivors has been analyzed in [47].

Tanimu et al. [48] conducted a research study with a primary focus on identifying risk factors linked to cervical cancer. They employed the decision tree (DT) classification algorithm and leveraged LASSO (least absolute shrinkage and selection operator) feature engineering methods and feature reduction techniques. These methods were used to pinpoint the critical characteristics to identify cervical cancer. The dataset utilized in their research presented challenges such as missing values and significant class imbalance. To tackle these issues, the research team adopted a technique known as SMOTETomek. The outcomes revealed that their proposed approach achieved outstanding accuracy. In a comparative analysis, Quinlan and colleagues [49] evaluated several ML classifiers for cervical cancer categorization. The dataset utilized in their analysis also showed signs of class imbalance, necessitating a method to solve this problem. To address the issue of class imbalance, the researchers used the SMOTE-Tomek in conjunction with a highly tailored RF. The findings showed that when combined with SMOTE-Tomek, the RF classifier attained an extraordinary accuracy rate.

Abdoh and colleagues [50] introduced a system for cervical cancer classification, employing the RF in conjunction with the SMOTE. hey also incorporated two feature reduction techniques. Their experiment used a dataset with thirty characteristics. The study looked at the effect of changing the feature size and discovered that utilizing SMOTE in conjunction with RF and other characteristics produced an excellent accuracy rate. Ijaz and colleagues [51] presented a data-driven approach for the detection of cervical cancer. The solution included both outlier identification and the SMOTE. The challenge was carried out utilizing the RF method with DBSCAN. According to their findings, when applied to a dataset of various features, their proposed method got a reasonable accuracy score.

In another work, Jahan et al. [52] developed an approach for detecting cervical cancer. Their study emphasized comparing the effectiveness of various classifiers in detecting the illness. The study included selecting multiple feature sets from the dataset and addressing missing data values using a mix of feature reduction approaches such as SelectBest, Chisquare, and RF. When applied to the top characteristics, the MLP algorithm achieved a remarkable accuracy rate. Mudawi et al. [53] proposed a complete research method for the prediction of cervical cancer that consists of four phases. They used a variety of ML classifiers in their research. According to the data, SVM achieved a phenomenal accuracy score in the cancer prediction job.

Following a thorough review of the literature, it is clear that multiple existing approaches have shown promising results in determining cervical cancer using diverse datasets. However, researchers have used a variety of optimisation methodologies to improve performance indicators including accuracy, precision, and recall. The major purpose of this work is to compare several ML algorithms in order to determine the best way to predict cervical cancer.

## Materials and methods

This section provides a quick yet thorough overview of some essential areas of cervical cancer screening. It covers the fundamentals, such as an introduction to the dataset, the painstaking processes used for data preparation, the ML algorithms used to predict cervical cancer, and the approaches used to address the difficulty of class imbalance in this context.

### Dataset

This study made use of a publicly accessible dataset from Venezuela's Hospital Universitario de Caracas [54]. This is the only publicly accessible dataset appropriate for a thorough study of cervical cancer screening utilising questionnaires and AI techniques. On this dataset, the researchers assessed the suitability and efficacy of AI models and data-balancing strategies.

Fig 1 presents a description of the dataset, which includes 858 instances and 36 characteristics. The table shows details about the dataset's input variables (35) and output variables (1). Fig 1 has a full explanation of each input variable. Notably, the dataset includes a variable called "Biopsy." There is a large class imbalance in the dataset. Recognising the inherent difficulties of categorising unbalanced data, the researchers chose to resolve missing values by oversampling the minority class using the SVM imputer approach and the ADASYN technique.

### Data preparation

Data preparation is critical for optimising the performance of machine learning models. It entails deleting unneeded or superfluous data, which can confound the models and reduce their efficiency. The data provided in Fig 1 highlights two primary concerns within the dataset.

- Missing Values

- Class Imbalance

**Handling missing values.** During data preprocessing in this study, it was observed that the dataset contained numerous missing data values. Fig 1 shows how the missing data values are distributed across different classes. Missing values are handled by removing them or by applying any imputation technique. This study employs an SVM Imputer to handle missing values.

| Number | Attribute name | Type | Range | Missing Values | % of missing values |
|--------|----------------|------|-------|----------------|---------------------|
| 1 | Age | int | 13–84 | 0 | 0% |
| 2 | IUD (Years) | int | 0–19 | 117 | 13.6% |
| 3 | STDs: genital herpes | bool | 0–1 | 105 | 12.2% |
| 4 | Harmonal contraceptives | bool | 0–1 | 108 | 12.5% |
| 5 | Dx: cancer | Bool | 0–1 | 0 | 0% |
| 6 | Smokes | Bool | 0–1 | 13 | 1.5% |
| 7 | STDs: vaginal condylomatosis | Bool | 0–1 | 105 | 12.2% |
| 8 | STDs: AIDS | Bool | 0–1 | 105 | 12.2% |
| 9 | Num of Pregnancies | Int | 0–110 | 56 | 6.5% |
| 10 | Intrauterine Device (IUD) | Bool | 0–1 | 117 | 13.6% |
| 11 | STDs: cervical condylomatosis | Bool | 0–1 | 105 | 12.2% |
| 12 | STDs: molluscum contagiosum | Bool | 0–1 | 105 | 12.2% |
| 13 | STDs: time since last diagnosis | Int | 0–3 | 787 | 91.7% |
| 14 | Cytology | Bool | 0–1 | 0 | 0% |
| 15 | First sex intercourse(age) | Int | 10–32 | 7 | 0.08% |
| 16 | Hormonal contraceptives (years) | Int | 0–22 | 108 | 12.5% |
| 17 | STDs: condylomatosis | Bool | 0–1 | 105 | 12.2% |
| 18 | STDs: Time since first diagnosis | Int | 0–1 | 787 | 91.7% |
| 19 | Schiller | Bool | 0–1 | 0 | 0% |
| 20 | Number of sexual partners | Int | 1–28 | 26 | 2.6% |
| 21 | Smokes (packs/year) | int | 0–37 | 13 | 1.5% |
| 22 | STDs (number) | Int | 0–4 | 105 | 12.2% |
| 23 | STDs: pelvic inflammatory diease | Bool | 0–1 | 105 | 12.2% |
| 24 | STDs: Number of diagnosis | Int | 0–1 | 0 | 0% |
| 25 | Hinselmann | Bool | 0–1 | 0 | 0% |
| 26 | Diagnosis: Dx | Bool | 0–1 | 0 | 0% |
| 27 | STDs: Hepatitis B | Bool | 0–1 | 105 | 12.2% |
| 28 | Smokes (years) | int | 0–37 | 13 | 1.5% |
| 29 | Sexually Transmitted Disease (STD) | Bool | 0–1 | 105 | 12.2% |
| 30 | STDs: syphilis | Bool | 0–1 | 105 | 12.2% |
| 31 | Dx: Human Papillomavirus (HPV) | Bool | 0–1 | 0 | 0% |
| 32 | STDs: vulvo-perineal condylomatosis | Bool | 0–1 | 105 | 12.2% |
| 33 | STDs: HPV | Bool | 0–1 | 105 | 12.2% |
| 34 | Dx: cervical intraepithelial Neoplasia (CIN) | Bool | 0–1 | 0 | 0% |
| 35 | STDs: HIV | Bool | 0–1 | 105 | 12.2% |
| 36 | Biopsy (target Variable) | bool | 0–1 | | |

**Fig 1. The description of dataset.**

SVM imputer [55] is an ML-based model that can be utilized to impute missing values in a dataset. It works by training an SVM classifier to estimate the absent values using the available data within the dataset. SVM imputer is particularly well-suited for imputing missing values in categorical data. This is because SVMs are able to learn complex relationships between categorical features. Here is a step-by-step overview of how SVM imputer works:

- Split the dataset into two parts: training and testing.

- Train an SVM classifier on the training set, using the known values to predict the missing data values.

- Use the trained SVM classifier to forecast the missing data values in the testing set.

SVM imputer can also be used to impute missing values in numerical data. However, it is important to note that SVM imputer is not as good at imputing missing values in numerical data as other imputation methods, such as mean imputation or median imputation.

**Handling class imbalance.** Class imbalance in datasets emerges when one group considerably outnumbers others, providing issues for ML models by biassing them towards the majority class [56]. This imbalance can lead to worse detection of minority groups, for as when identifying illnesses like cancer, affecting model accuracy and patient outcomes. Addressing this imbalance is vital for ensuring fair and accurate learning, eliminating biases towards majority classes, and boosting the model's capacity to recognise all classes successfully, especially in critical areas like medical diagnosis.

To address the dataset's class imbalance issue, ADASYN [57] is implemented. ADASYN (Adaptive Synthetic Sampling) is an ML technique used to address class imbalance in datasets. Class imbalance occurs when there is a significant difference in the sample size from different classes in a dataset. This can be a problem for machine learning models, as they may learn to focus on the majority class and neglect the minority class.

ADASYN works by generating synthetic samples for the minority class. The amount of synthetic samples created for each instance of a minority class is determined by the difficulty of learning that instance. Instances that are more difficult to learn are assigned more synthetic samples. This helps to make the dataset balance and enhance the efficacy of ML models on the minority class. ADASYN is a reasonably basic method that has been demonstrated to improve the performance of ML models on unbalanced datasets.

## Feature extraction

The CNN (Convolutional Neural Network) model is used in this study for feature engineering in the detection of cervical cancer. The CNN model, like other deep learning models, has many layers: the embedding, the max-pooling, and the convolutional layer.

The initial layer, referred to as the embedding layer, utilizes all attributes from the cervical cancer dataset, employing an embedding size of 25,000 and producing an output with a dimensionality of 300. Following the embedding layer is a Conv-1D layer containing 4,000 filters. This layer incorporates the ReLU (Rectified Linear Unit) activation function and employs a 2x2 kernel size. To capture pertinent features, the output of the 1D convolution is subjected to a 2x2 max-pooling layer. Lastly, a flatten layer is applied to convert the output back into a 1D array, ensuring compatibility with the ML model.

The cervical cancer dataset is structured as a tuple set ($fs_i$, $tc_i$), with $fs$ representing the feature set, $tc$ indicating the target class column, and $i$ denoting the tuple index. The embedding layer is employed to transform the training set into the intended input format in the following manner:

$$EL = embeddinglayer(Vs, Os, I) \qquad (1)$$

$$EOs = EL(fs) \qquad (2)$$

$EO_s$ represents the output generated by the embedding layer, serving as the input for the subsequent convolutional layer. The embedding layer's parameters encompass the input lengths ($I$), vocabulary size ($Vs$), and output dimensions ($Os$).

## Machine learning classifiers

This section gives an in-depth look at the machine learning methods utilised in this study, including implementation details. These algorithms are implemented using the scikit-learn and NLTK libraries. This job employs eight supervised ML techniques that are typically used for classification and regression problems. These algorithms were written in Python and implemented using the scikit-learn module.

XGBoost [58], or Extreme Gradient Boosting, is a powerful ensemble machine learning model based on gradient boosting. It excels in predictive modelling tasks by combining multiple decision trees in a way that corrects errors sequentially, leading to a robust ensemble model. XGBoost uses regularized tree-building algorithms to prevent overfitting and optimize model complexity. It is known for its efficiency, scalability, and versatility, making it suitable for various ML tasks. Features like built-in handling of missing values, feature importance analysis, and a supportive community contribute to its popularity. Careful hyperparameter tuning is essential to maximize its performance in specific applications. XGBoost has acquired robust results in ML competitions and is widely used in real-world applications.

Random Forest [59] is a versatile ensemble ML algorithm that combines multiple decision trees to make predictions. It employs a bagging technique to reduce overfitting and bias, making it robust and effective. Random Forest is known for its feature importance analysis, scalability, and versatility in handling various types of data. It's widely used in classification and regression tasks due to its strong performance and ability to handle complex datasets.

The Stochastic Gradient Descent (SGD) [60] is an optimization algorithm commonly used for linear classification tasks. It incorporates randomness by selecting one training example at a time during each iteration, making it computationally efficient, especially for large datasets and online learning scenarios. SGD is versatile and can be employed with various regularization techniques. It's valued for its scalability and suitability for real-time applications but requires careful tuning of hyperparameters for optimal performance.

K-Nearest Neighbors (KNN) [61] is an ML algorithm used for classification and regression tasks. It operates on the principle of finding the K nearest data points in the training dataset to make predictions for new data points. Key characteristics include its non-parametric nature, reliance on instance-based learning, and sensitivity to the choice of the hyperparameter K. KNN is versatile, used in various applications, and robust to outliers, but it can be affected by the curse of dimensionality in high-dimensional spaces. It's a simple yet effective algorithm for both linear and non-linear data distributions.

Logistic Regression [62] is a versatile ML algorithm primarily used for binary categorization tasks. It predicts the probability of an input belonging to one of two classes (typically 0 and 1) by applying a logistic (sigmoid) function to ensure output values between 0 and 1. This algorithm is known for its simplicity, transparency, and interpretability, making it valuable in scenarios where understanding the impact of each input feature is crucial. However, it may not perform well with highly nonlinear relationships between features and outcomes, and it assumes independence among features.

The Extra Tree Classifier [63] is an ensemble machine learning algorithm that combines multiple decision trees. It stands out due to its high level of randomization during tree construction, reducing the risk of overfitting and making it robust against noisy data. Like Random Forest, it employs bootstrapping and provides feature importance scores. The Extra Tree Classifier is versatile, scalable, and applicable to both classification and regression tasks. Careful hyperparameter tuning is necessary for optimal performance, though it's generally less sensitive to hyperparameters compared to some other algorithms.

**Proposed approach.** The study made use of a dataset obtained from Kaggle, a recognised platform for publicly available datasets. A range of preprocessing processes were carried out to improve the efficacy of ML models and handle missing data. The SVM imputer was used to deal with missing data values. Following that, the dataset was divided into a 70:30 ratio, with 70% allotted as the train set and 30% as a test set. The suggested system used an ensemble technique known as RF + KNN + LR to identify cervical cancer. Voting classifiers are powerful strategies that aggregate results from many models to improve accuracy and durability. The

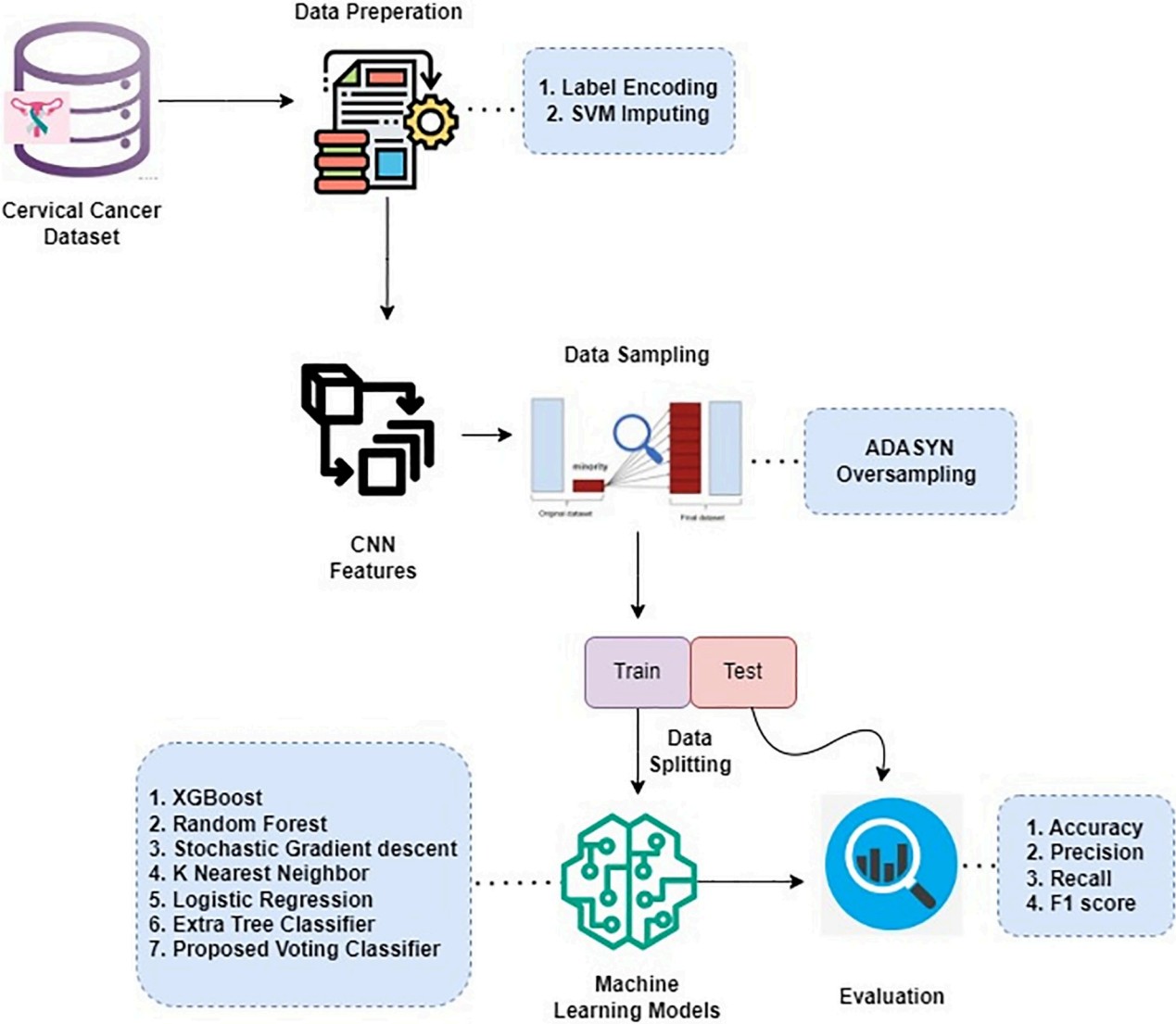

**Fig 2. The workflow of the proposed methodology.**

models in the voting classifier have their own set of strengths and shortcomings, and their combined use results in greater overall performance. In this scenario, the suggested method for detecting cervical cancer incorporates three commonly used algorithms: RF, KNN, and LR. Fig 2 shows a process diagram explaining this strategy.

The voting classifier works by combining predictions from these three different ML methods. The typical strategy for building an ensemble/voting classifier is training numerous classifiers using the dataset and then combining their results. In this case, the RF, KNN, and LR models were all trained individually on the same dataset. Each model predicts the probability of each class inside the target variable. The estimated probabilities are then summed to get a final forecast for each instance in the dataset. A popular method for integrating the predictions is to compute a weighted average of the expected probabilities defined by the performance of each model on a validation dataset.

**Algorithm 1** Ensembling of SV-CNN model.
**Input:** input data $(x, y)_{i=1}^N$
$M_{RF}$ = Trained_RF
$M_{KNN}$ = Trained_KNN
$M_{LR}$ = Trained_LR
1: **for** $i = 1$ $to$ $M$ **do**
2: **if** $M_{RF} \neq 0$ & $M_{KNN} \neq 0$ & $M_{LR} \neq 0$ & $training\_set \neq 0$ **then**
3: $ProbRF - 1 = M_{RF}.probability(1 - class)$
4: $ProbRF - 2 = M_{RF}.probability(2 - class)$
5: $ProbCNN - 1 = M_{KNN}.probability(1 - class)$
6: $ProbCNN - 2 = M_{KNN}.probability(2 - class)$
7: $ProbCNN - 1 = M_{LR}.probability(1 - class)$
8: $ProbCNN - 2 = M_{LR}.probability(2 - class)$
9: Decision function = $max\left(\frac{1}{N_{classifier}}\sum_{classifier}\right)$
 $(Avg_{(ProbRF-1, \ ProbKNN-1, \ ProbLR-1)}$
 , $(Avg_{(ProbRF-2, \ ProbKNN-2, \ ProbLR-2)}$
10: **end if**
11: Return final label $\hat{p}$
12: **end for**

Lines 3 to 6 of algorithm 1 indicate the probability scores of classes 1 and 2 from RF, KNN, and LR models, respectively. The probability score of a classifier, often used in classification tasks, represents the likelihood or confidence that a given data sample belongs to a particular class. It is calculated based on the output of the classifier model, such as in this case RF, KNN, and LR. This probability is not the final prediction of a specific target class. It is like a raw score (prediction confidence). To convert raw scores into probabilities, some classifiers employ a probability calibration step. This step ensures that the calculated scores are well-calibrated and can be interpreted as probabilities. Based on the probability scores and the chosen threshold (0.5 in this case), the classifier assigns a final predicted class label to the data sample. The decision function of line 7 of algorithm 1, decides the final class based on the class which has more probability score than the assigned threshold. The working example of the decision function is added below for further clarification.

The working of this ensemble can be explained using an example. Each sample that undergoes processing by both the SVM and CNN is assigned a probability score. Consider a scenario where the RF model assigns a probability of 0.4 and 0.7 for class 1 and class 2, KNN model assigns a probability of 0.5 and 0.8 for class 1 and class 2, respectively, and the LR model assigns probability scores of 0.5 and 0.4 for the same two classes. Denoting the probability value of $x$ as $P(x)$, where $x$ ranges from 1 to 2, the final probability is calculated as follows:

$P(1) = (0.4 + 0.5 + 0.5)/3 = 0.46$

$P(2) = (0.7 + 0.8 + 0.4)/3 = 0.63$

This ensemble approach makes the final class label based on the probability scores for each class from both models used for voting. The final label is decided based on the highest average probability using line 7 of algorithm 1.

The proposed model uses the distinct capabilities of three separate ML models to provide results that are both accurate and resilient. To generalize the model while decreasing the overfitting by training models on the cervical cancer dataset and fusing their results of predictions. The suggested ensemble model's fundamental functionality may be summarised as follows:

$$\hat{p} = argmax\{\sum_i^n RF_i, \sum_i^n KNN_i, \sum_i^n LR_i\}. \tag{3}$$

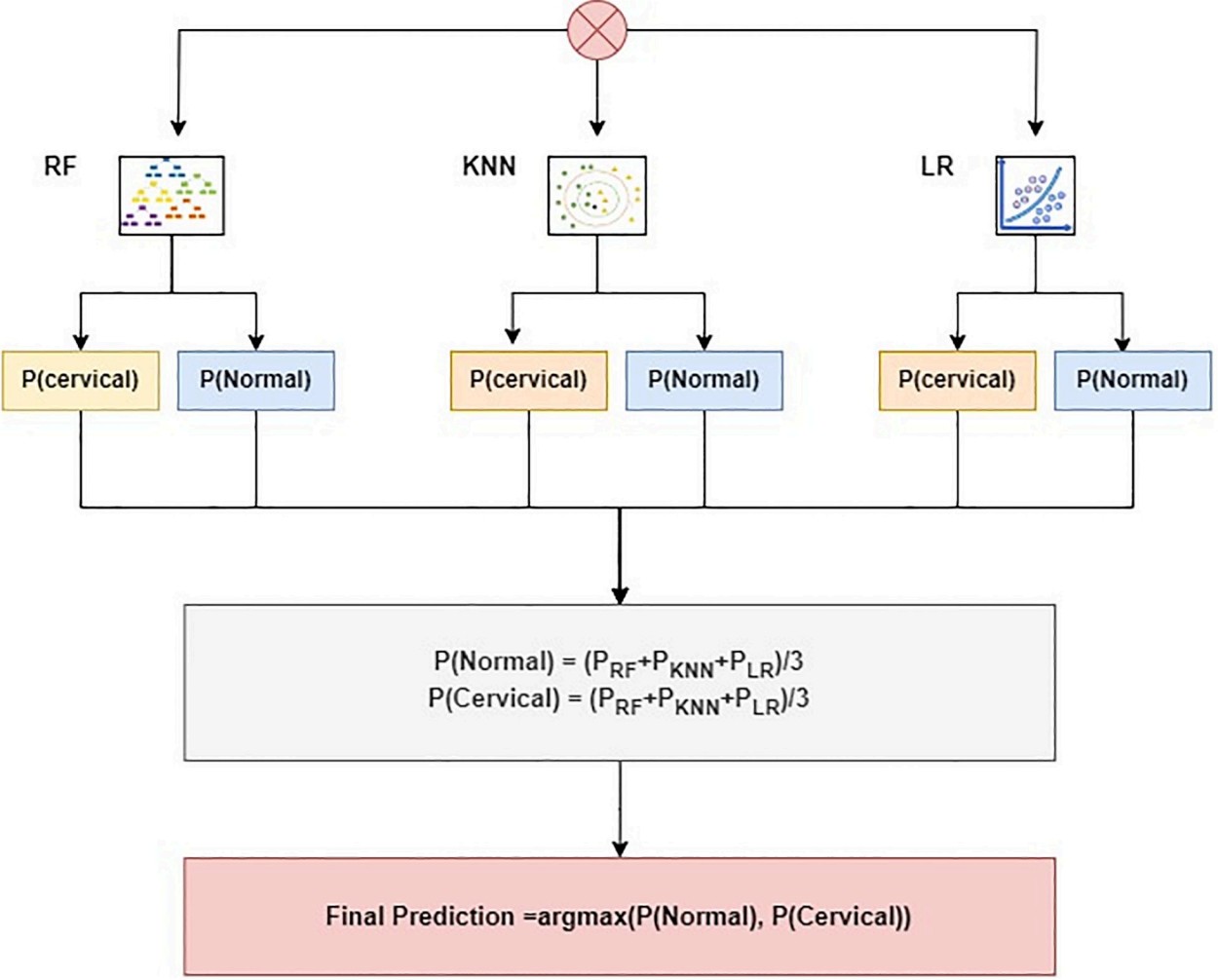

**Fig 3. Structure of the proposed ensemble voting classifier.**

where $\sum_{i}^{n} RF_i$, $\sum_{i}^{n} KNN_i$, and $\sum_{i}^{n} LR_i$ indicate the prediction probability for each test sample *RF*, *KNN*, and *LR*, respectively. After that, each test case's probabilities acquired from *RF*, *KNN*, and *LR* are subjected to the soft voting criterion, as shown in Fig 3.

The proposed voting classifier identifies the optimal class prediction by evaluating the class with the highest average probability across all classes. This is achieved by amalgamating the projected probabilities from both models. The final prediction is made by selecting the class with the highest score of probability, as illustrated below:

$$VC(RF + KNN + LR) = argmax(g(x)) \qquad (4)$$

### Evaluation metrics

The proposed system generates four crucial indices for evaluating its performance: True Negative (TN), True Positive (TP), False Negative (FN), and False Positive (FP). The evaluation metric used in this study to evaluate the models is presented in Table 1.

**Table 1. Assessment metrics for evaluating performance.**

| Evaluation Metric | Formula |
|---|---|
| Accuracy (A) | $\frac{TP+TN}{TP+TN+FP+FN}$ |
| Precision(P) | $\frac{TP}{TP+FP}$ |
| Recall (R) | $\frac{TP}{TP+FN}$ |
| F1 Score (F) | $2*\frac{precision.recall}{precision+recall}$ |

## Results

This section discusses the results of experiments and their consequences, with an emphasis on determining the efficacy of the suggested methodology in comparison to existing methodologies. The assessment includes a variety of test parameters used for the cervical cancer dataset, and the results are contrasted with alternative ML approaches. The tests are conducted using the original dataset, the SVM Imputed dataset, the dataset upsampled using AdDASYN and imputed with SVM, and the dataset containing CNN features.

### Classifier performance with the original dataset

Initially, the experiments are conducted using the original dataset extracted from the cervical cancer dataset. The outcomes of all ML models utilizing these original features are presented in Table 2.

The findings demonstrate that LR and KNN had the best levels of accuracy among the classifiers, with rates of 73.41% and 72.98%, respectively. The precision of RF was 78.35%, the recall was 79.95%, and the F1 score was 79.91%. KNN exhibited precision and recall of 81.45%, resulting in an F1 score of 81.45%. Similarly, LR attained a precision of 80.15%, recall of 80.12%, and F1 score of 80.17%. XGB, on the other hand, performed the least successfully, with an accuracy rate of 64.57%, precision of 77.54%, recall of 79.64%, and F1 score of 78.51%.

The proposed Voting Classifier (VC), which merged RF, KNN, and LR, demonstrated superiority in terms of performance when compared to all individual models. It achieved an accuracy of 80.13%, precision of 84.56%, recall of 86.31%, and an F1 score of 85.71%. Nevertheless, when the individual machine learning models were assessed using the dataset without any missing values, their performance frequently lagged behind.

### Classifier performance with SVM imputed dataset

The SVM imputer was used in the following phase of the trials to handle missing values in the dataset. Some values were missing during the data preparation step, forcing the employment

**Table 2. Results of the machine learning models obtained using the original dataset.**

| Model | A(%) | P(%) | R(%) | F(%) |
|---|---|---|---|---|
| XGB | 64.57 | 77.54 | 79.64 | 78.51 |
| RF | 71.55 | 78.35 | 79.95 | 79.91 |
| SGD | 68.49 | 76.27 | 78.78 | 77.56 |
| KNN | 72.98 | 81.45 | 81.45 | 81.45 |
| LR | 73.41 | 80.15 | 80.12 | 80.17 |
| ETC | 69.25 | 76.24 | 81.34 | 78.52 |
| VC (RF + KNN + LR) | 80.13 | 84.56 | 83.31 | 85.71 |

**Table 3. Results of the learning models using SVM imputer.**

| Model | A(%) | P(%) | R(%) | F(%) |
|---|---|---|---|---|
| XGB | 73.57 | 86.54 | 88.64 | 87.51 |
| RF | 81.65 | 89.35 | 90.88 | 90.31 |
| SGD | 78.69 | 86.41 | 88.83 | 87.86 |
| KNN | 83.10 | 90.33 | 90.33 | 90.33 |
| LR | 84.72 | 89.74 | 90.25 | 90.01 |
| ETC | 80.54 | 88.42 | 89.43 | 89.25 |
| VC (RF + KNN + LR) | 97.41 | 97.63 | 95.96 | 96.76 |

of the SVM imputer to bridge these gaps. Following the imputation procedure, the amended dataset was used to train and evaluate several ML models. Table 3 describes the findings of these models.

According to the results, KNN, and LR attained accuracy rates of 83.10%, and 84.72%, respectively. However, the suggested VC (RF+KNN+LR) greatly beat them all, with an outstanding accuracy rate of 97.41%.

## Classifier performance with ADASYN upsampled

The ADASYN approach was used in the third round of studies to address the class imbalance problem in the dataset. In the data preprocessing step, it became clear that only 58 of the total 858 samples were from the malignant class. To address the issue of class imbalance, ADASYN was utilized as an oversampling strategy. The enhanced dataset was utilised to evaluate the performance of several ML models. The results of these models is summarised in Table 4.

The results show that the suggested voting ensemble model VC(RF + KNN + LR) surpasses all other models with an amazing accuracy of 94.24%. Individual classifiers LT, RF, and KNN all earned outstanding accuracy ratings of 85.37%, 83.48%, and 84.19%, respectively. Nonetheless, the VC ensemble of linear models (RF + KNN + LR) outperformed the other models on the up-sampled dataset.

## Classifier performance with CNN generated features

The results from the fourth series of experiments, which involved utilizing the CNN-generated features with the SVM imputer for missing value handling and ADASYN for addressing class imbalance, can be found in Table 5. By leveraging both the CNN and ADASYN together, the objective was to simultaneously address missing values and class imbalance, aiming to improve the accuracy of the linear model. Following the application of the SVM imputer and ADASYN, ML models were trained and evaluated.

**Table 4. Results of the learning models using ADASYN upsampled.**

| Model | A(%) | P(%) | R(%) | F(%) |
|---|---|---|---|---|
| XGB | 76.57 | 80.54 | 81.34 | 80.22 |
| RF | 83.48 | 85.34 | 86.34 | 86.00 |
| SGD | 79.58 | 82.54 | 83.27 | 83.08 |
| KNN | 84.19 | 87.24 | 88.92 | 88.24 |
| LR | 85.37 | 87.85 | 88.68 | 88.31 |
| ETC | 77.67 | 78.34 | 79.74 | 78.37 |
| VC (RF + KNN + LR) | 94.24 | 94.89 | 95.19 | 95.06 |

**Table 5. Results of machine learning models with CNN generated features.**

| Model | A(%) | P(%) | R(%) | F(%) |
|---|---|---|---|---|
| XGB | 96.58 | 98.18 | 99.35 | 99.45 |
| RF | 93.55 | 95.19 | 96.49 | 96.19 |
| SGD | 98.89 | 92.22 | 93.19 | 93.84 |
| KNN | 97.91 | 97.38 | 98.35 | 98.37 |
| LR | 95.68 | 97.67 | 98.88 | 97.19 |
| ETC | 97.79 | 98.49 | 99.49 | 98.93 |
| VC (RF + KNN + LR) | 99.99 | 99.99 | 99.99 | 99.99 |

**Table 6. Significance of proposed methodology using k-fold validation.**

| Fold Number | A(%) | P(%) | R(%) | F(%) |
|---|---|---|---|---|
| Fold-1 | 99.23 | 99.96 | 99.94 | 99.95 |
| Fold-2 | 99.34 | 99.96 | 99.95 | 99.96 |
| Fold-3 | 99.45 | 99.97 | 99.96 | 99.96 |
| Fold-4 | 99.11 | 99.94 | 100.0 | 99.99 |
| Fold-5 | 99.24 | 99.99 | 99.98 | 99.99 |
| **Average** | **99.27** | **99.96** | **99.96** | **99.97** |

## Results of cross-validation

A 5-fold cross-validation was undertaken to further confirm the efficacy of the suggested technique. The findings are shown in Table 6. Notably, the suggested model has an average accuracy of 99.27, as well as average precision, recall, and F1 score values of 99.96%, 99.96%, and 99.97%.

## Discussions

The comparison of classifiers with the original dataset, with SVM Imputed dataset, With ADASYN upsampled and SVM Imputed dataset, and with CNN generated features dataset is depicted in Fig 4. When applied to ADASYN-balanced data, the Voting classifier demonstrates its superiority over all other classifiers. RF, KNN and LR perform better than other individual classifiers in every scenario. These findings underscore the significance of selecting the right combination of ML approaches for the effective operation of an ensemble ML model. In the context of analyzing imbalanced text data, employing data balancing techniques like ADASYN significantly enhances classifier performance.

This highlights the importance of utilizing the statistical technique ADASYN to balance the data before training, as it plays a pivotal role in enhancing classifier performance. It becomes apparent that classifiers may not achieve their optimal performance when dealing with imbalanced classes within the original dataset. Consequently, when coupled with the SVM imputer and the ADASYN technique for cervical cancer detection, the proposed model demonstrates improved generalization and outperforms other models when appropriately configured.

ADASYN and SVM imputer are very useful techniques in improving the performance of the models with class imbalance problems. When experiments are performed using CNN features, the results of the classifier have shown significant improvement in results. The proposed voting classifier has outperformed with a 99.99% score of accuracy, precision, recall and F1 score. Ensembling RF, KNN, and LR models offer a strategic advantage by combining their

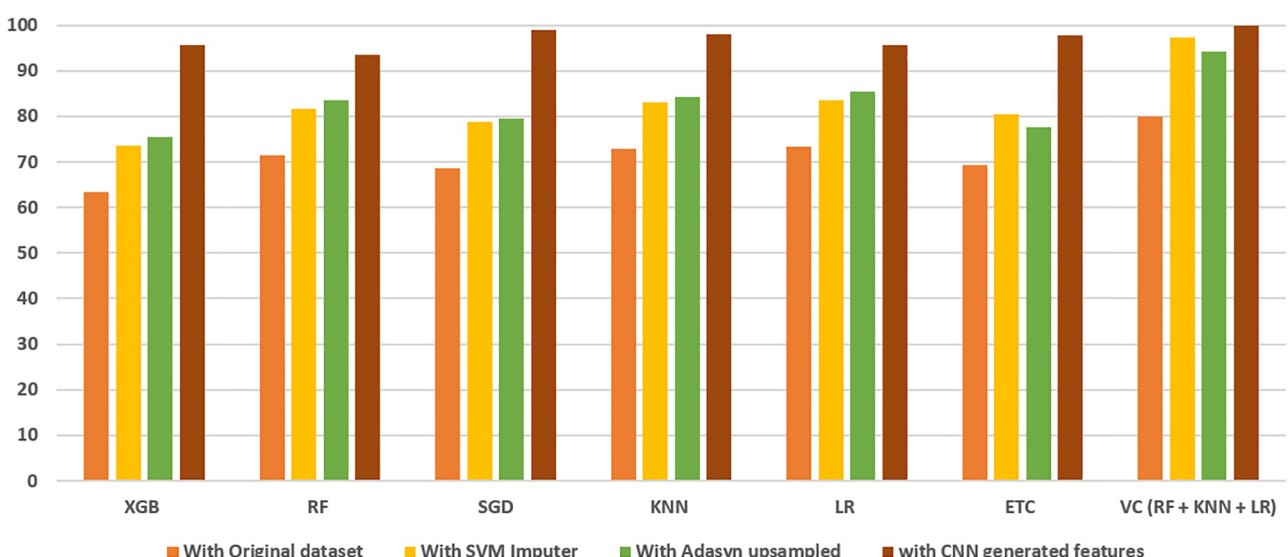

**Fig 4. Comparison of model accuracies in all scenarios.**

diverse learning approaches to improve predictive performance. This ensemble aims to leverage the strengths of each model: RF's robustness, KNN's pattern recognition, and LR's probabilistic interpretation, thereby enhancing overall accuracy, reducing overfitting, improving robustness against outliers, and providing a more comprehensive analysis of the cervical cancer dataset. The ensemble, facilitated by a voting classifier, fosters a collective decision-making process, resulting in a more robust and accurate predictive model for cervical cancer detection.

## Comparative analysis with cutting-edge methods

To assess the efficacy of the suggested method, a performance comparison with existing models specialised in cervical cancer diagnosis is performed. This review includes a selection of current research from the existing studies that serve as comparative points. According to one research [48], a cancer detection model using Recursive Feature Elimination (RFE) and DT using SMOTETomek obtains an accuracy of 98.82%, precision of 87.53%, recall of 100%, and an F1 score of 93.333%. Another research [51] uses 10 features for the same job and obtains an accuracy score of 97.72%. Furthermore, studies [52] applied MLP and [53] used SVM and revealed accuracy rates of 98.10% and 99%, respectively.

Despite the great accuracy reported in previous research works, the suggested models outperform them, as shown in Fig 5. The suggested method outperforms previous approaches due to three main factors: managing missing data, employing an ensemble voting classifier, and adding CNN-generated features. The novel mix of strategies, which includes correcting missing data, applying ensemble learning, and regulating class imbalance, is critical to the observed accuracy gains. While some earlier techniques may have failed to resolve the problem of missing data directly, this work uses an SVM imputation strategy in conjunction with ADASYN up-sampled features. In addition, the suggested technique makes use of a stacked ensemble voting classifier, which combines the results of three independent models. This collaborative approach is advantageous.

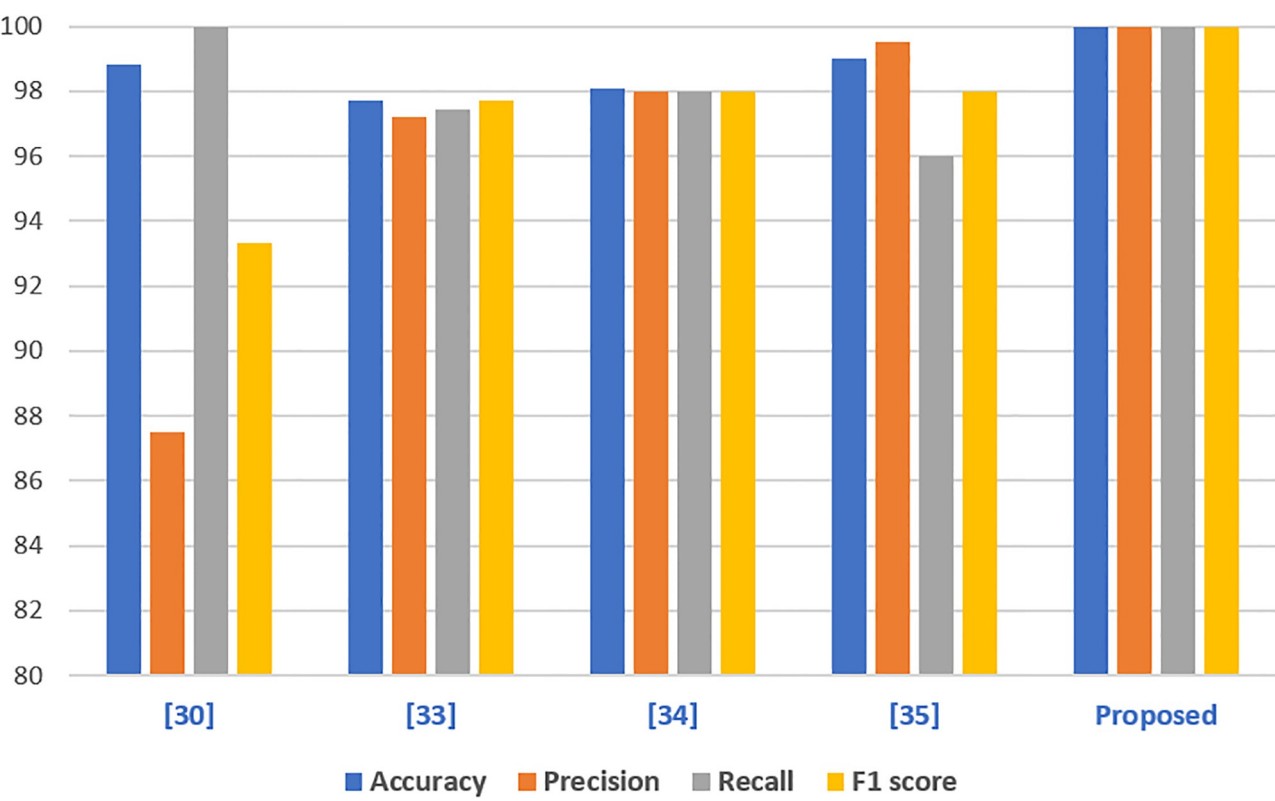

**Fig 5. Comparison of model accuracies in all scenarios.**

## Conclusions

Cervical cancer poses a significant threat to women's health, particularly in developing countries, where it ranks as a leading cause of mortality. Timely detection and treatment, guided by skilled medical professionals, are paramount in mitigating its devastating impact. Pap smear images have emerged as valuable diagnostic tools for identifying this form of cancer. However, numerous datasets designed for automated cervical cancer detection present a common challenge: missing values. These gaps in data can substantially hinder the performance of machine learning models, necessitating innovative solutions.

In response to these challenges, this study introduces an automated system tailored for cervical cancer prediction. This system demonstrates remarkable proficiency in managing missing values through the utilization of ADASYN features, ultimately achieving exceptional levels of accuracy. The cornerstone of the proposed approach is a stacked ensemble voting classifier model, strategically combining the predictive capabilities of three distinct machine learning models. Furthermore, SVM Imputer and ADASYN up-sampled features are integrated into the proposed framework to effectively address concerns related to missing values. The inclusion of CNN-generated features further bolsters the model's robustness.

Notably, the outcomes of this study reveal the exceptional performance of the proposed model, boasting remarkable metrics such as 99.99% accuracy, 99.99% precision, 99.99% recall, and a 99.99% F1 score. To comprehensively assess the proposed model, a comparative analysis is conducted against various machine learning algorithms under four distinct scenarios: using the original dataset, employing SVM imputation, incorporating ADASYN features, and

harnessing CNN-generated features. These comparative evaluations underscore the superior efficacy of the proposed model when compared to existing state-of-the-art approaches. The research has the potential to significantly benefit medical practitioners by enabling earlier cervical cancer detection and improving patient care. Future work aims to develop stacked ensembles of machine and deep learning models for enhanced performance on higher-dimensional datasets.

## Supporting information

**S1 File.**
(ZIP)

## Author Contributions

**Conceptualization:** Raafat M. Munshi.

**Data curation:** Raafat M. Munshi.

**Formal analysis:** Raafat M. Munshi.

**Funding acquisition:** Raafat M. Munshi.

**Investigation:** Raafat M. Munshi.

**Methodology:** Raafat M. Munshi.

**Project administration:** Raafat M. Munshi.

**Resources:** Raafat M. Munshi.

**Software:** Raafat M. Munshi.

**Supervision:** Raafat M. Munshi.

**Validation:** Raafat M. Munshi.

**Visualization:** Raafat M. Munshi.

**Writing – original draft:** Raafat M. Munshi.

**Writing – review & editing:** Raafat M. Munshi.

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
