## [Decision Letter · Decision Letter 0]

17 Oct 2023

PONE-D-23-31768Novel Ensemble Learning Approach with SVM-Imputed ADASYN Features for Enhanced Cervical Cancer PredictionPLOS ONE

Dear Dr. Munshi,

Thank you for submitting your manuscript to PLOS ONE. After careful consideration, we feel that it has merit but does not fully meet PLOS ONE’s publication criteria as it currently stands. Therefore, we invite you to submit a revised version of the manuscript that addresses the points raised during the review process.

We look forward to receiving your revised manuscript.

Kind regards,

Muhammad Umer, Ph.D. in Computer Science

Academic Editor

PLOS ONE

Journal Requirements:

Additional Editor Comments:

The reviewers suggested that the paper has potential but requires major revision to improve the quality. Address the changes asked by reviewers. Also improve the English editings and typos.

Reviewers' comments:

Reviewer's Responses to Questions

**Comments to the Author**

1. Is the manuscript technically sound, and do the data support the conclusions?

Reviewer #1: Yes

Reviewer #2: Yes

2. Has the statistical analysis been performed appropriately and rigorously? 

Reviewer #1: Yes

Reviewer #2: N/A

3. Have the authors made all data underlying the findings in their manuscript fully available?

Reviewer #1: Yes

Reviewer #2: Yes

4. Is the manuscript presented in an intelligible fashion and written in standard English?

Reviewer #1: Yes

Reviewer #2: No

5. Review Comments to the Author

Reviewer #1: The paper is interesting from medical point of view. But the paper has some major lacks in current state which needs to be addressed to make any final decision about it. There are a lot of grammatical mistakes as well as some claims/points needs to be more refined to attains the attention of the readers. Therefore, I suggest a major revision for this work.

Basic Reporting

1. Introduction section need revision, it is recommended to present one idea in one paragraph.

2. Add some recently published works for cervical cancer survival prediction in related work section.

3. Research gap is not clear.

4. What is the motivation and rationale of the study.

5. Figure quality is not good.

6. Empirical analysis of the proposed methodology is missing.

7. Authors should re-write the Abstract based on novelty, Challenges, validation techniques, and some suggestion. Impact on social and Clinical.

8. Stop using words like our work and we. Used proposed work or proposed model in the entire paper.

9. Stick to one term either F1 score or F1-score in the entire paper.

10. Finally, grammatical proofing also required to improve the interest of the readers.

Reviewer #2: First, I appreciate the efforts made by single author Raafat to design this research methodology and completed the projected. The theme of the paper is interesting but the paper still has a lot of flaws that needed to be addressed.

a. The introduction is okay but lacks application of CNN and bioinformatics state of the art research work. Author needed to cite some latest research work to cover up this gap.

b. Remove the terms authors written each time with first author name or use words like this research work.

c. Add some justification about why RF KNN and LR are ensembled.

d. Add an example to show how the voting classifier works for ensemble learning.

e. Explain with reference, what is class imbalance problem and why it is necessary to solve

f. Why Adasyn features, why not smote?

g. English editing is compulsory.

6. PLOS authors have the option to publish the peer review history of their article (what does this mean?). If published, this will include your full peer review and any attached files.

Reviewer #1: **Yes: **Dr. Omar I. Aboulola

Reviewer #2: No

---

## [Author Response · Author response to Decision Letter 0]

24 Nov 2023

We have added a separate PDF file for response to reviewers.

---

## [Decision Letter · Decision Letter 1]

6 Dec 2023

Novel Ensemble Learning Approach with SVM-Imputed ADASYN Features for Enhanced Cervical Cancer Prediction

PONE-D-23-31768R1

Dear Dr. Munshi,

We’re pleased to inform you that your manuscript has been judged scientifically suitable for publication and will be formally accepted for publication once it meets all outstanding technical requirements.

Kind regards,

Muhammad Umer, Ph.D. in Computer Science

Academic Editor

PLOS ONE

Additional Editor Comments:

No further comments. 

Reviewers' comments:

Reviewer's Responses to Questions

**Comments to the Author**

1. If the authors have adequately addressed your comments raised in a previous round of review and you feel that this manuscript is now acceptable for publication, you may indicate that here to bypass the “Comments to the Author” section, enter your conflict of interest statement in the “Confidential to Editor” section, and submit your "Accept" recommendation.

Reviewer #1: All comments have been addressed

Reviewer #2: All comments have been addressed

2. Is the manuscript technically sound, and do the data support the conclusions?

Reviewer #1: Yes

Reviewer #2: Yes

3. Has the statistical analysis been performed appropriately and rigorously? 

Reviewer #1: Yes

Reviewer #2: Yes

4. Have the authors made all data underlying the findings in their manuscript fully available?

Reviewer #1: Yes

Reviewer #2: Yes

5. Is the manuscript presented in an intelligible fashion and written in standard English?

Reviewer #1: Yes

Reviewer #2: Yes

6. Review Comments to the Author

Reviewer #1: Agree and accepted and all comments have been reviewed and has made better outcome. The paper has been pulished and the auther has met and change all required

Reviewer #2: all of my comments are resolved. The author did will in the revision, and paper quality improves a lot.

7. PLOS authors have the option to publish the peer review history of their article (what does this mean?). If published, this will include your full peer review and any attached files.

Reviewer #1: No

Reviewer #2: No

---

## [Editor Report · Acceptance letter]

28 Dec 2023

PONE-D-23-31768R1 

PLOS ONE

Dear Dr. Munshi, 

I'm pleased to inform you that your manuscript has been deemed suitable for publication in PLOS ONE. Congratulations! Your manuscript is now being handed over to our production team.

Kind regards, 

on behalf of

Dr. Muhammad Umer 

Academic Editor

PLOS ONE